# Infections, Reactions of Natural Killer T Cells and Natural Killer Cells, and Kidney Injury

**DOI:** 10.3390/ijms23010479

**Published:** 2022-01-01

**Authors:** Takahiro Uchida, Shuhji Seki, Takashi Oda

**Affiliations:** 1Kidney Disease Center, Department of Nephrology and Blood Purification, Tokyo Medical University Hachioji Medical Center, Tokyo 193-0998, Japan; tu05090224@gmail.com; 2Department of Immunology and Microbiology, National Defense Medical College, Saitama 359-8513, Japan; rui-21@jcom.home.ne.jp

**Keywords:** acute kidney injury, CD56, CD56^+^ T cell, infection, natural killer cell, natural killer T cell

## Abstract

Natural killer T (NKT) cells and NK cells are representative innate immune cells that perform antitumor and antimicrobial functions. The involvement of these cells in various renal diseases, including acute kidney injury (AKI), has recently become evident. Murine NKT cells are activated and cause AKI in response to various stimuli, such as their specific ligand, cytokines, and bacterial components. Both renal vascular endothelial cell injury (via the perforin-mediated pathway) and tubular epithelial cell injury (via the tumor necrosis factor-alpha/Fas ligand pathway) are independently involved in the pathogenesis of AKI. NK cells complement the functions of NKT cells, thereby contributing to the development of infection-associated AKI. Human CD56^+^ T cells, which are a functional counterpart of murine NKT cells, as well as a subpopulation of CD56^+^ NK cells, strongly damage intrinsic renal cells in vitro upon their activation, possibly through mechanisms similar to those in mice. These cells are also thought to be involved in the acute exacerbation of pre-existing glomerulonephritis triggered by infection in humans, and their roles in sepsis-associated AKI are currently under investigation. In this review, we will provide an overview of the recent advances in the understanding of the association among infections, NKT and NK cells, and kidney injury, which is much more profound than previously considered. The important role of liver macrophages in the activation of NKT cells will also be introduced.

## 1. Introduction

Infections are known to induce various types of kidney injury. One of the most common types is acute kidney injury (AKI) [1], which in turn causes the dysfunction of distant organs, such as the lung or heart [2]. Patients with AKI are known to have both high mortality and morbidity rates, and the development of AKI in patients with sepsis is associated with a further increase in mortality; i.e., more than half of patients with sepsis-associated AKI reportedly die [3]. AKI has also been recognized as an important cause of chronic kidney disease, which may progress to end-stage kidney disease (ESKD), and is a significant economic burden [2,4]. A better understanding of the pathogenesis of AKI and establishing effective therapeutic approaches to AKI are therefore urgent issues.

However, the pathogenesis of AKI is complicated, and involves proinflammatory cytokines, reactive oxygen species (ROS), changes in hemodynamics, and cell apoptosis [5]. Furthermore, the roles of dysregulation of the immune system in AKI have recently gained attention [6]. Both innate and adaptive immune responses are considered to be involved in the development of AKI.

Murine natural killer T (NKT) cells, which express the NK1.1 antigen and intermediate levels of the T-cell receptor (TCR) with a restricted variance (mainly Vα14Jα18/Vβ8), and have a phenotype of CD4 (or CD4^−^ CD8^−^), and NK cells are innate immune cells that are present abundantly in the liver [7,8]. NKT cells that express the invariant TCR are called invariant NKT cells. They have a morphology of large granular lymphocytes, and exist in peripheral tissues in the absence of inflammation, from early postnatal life or during fetal development [9]; and hence are completely different from conventional T cells [10]. The function of NK cells is controlled by an interplay of activating receptors, which detect upregulated stress molecules on damaged cells showing reduced or aberrant major histocompatibility complex (MHC) class I expression, and inhibitory receptors, which identify normal cells by recognizing their intact MHC class I expression, referred to as the recognition of missing self [11,12]. NK cells can also be activated directly by cytokine signals, and their cell-surface expression of receptors for the Fc portion of antibodies can mediate antibody-dependent cellular cytotoxicity [12]. As with NK cells, NKT cells express activating and/or inhibitory receptors and receptors for cytokines, such as interleukin (IL)-12 and tumor necrosis factor-alpha (TNF-α) [13,14]. However, NKT cells can also be activated via TCR ligation; once activated by an anti-CD3 antibody, they show strong cytotoxic functions [7]. Furthermore, their specific sphingoglycolipid ligand that is presented on CD1d molecules on macrophages activates NKT cells to produce large amounts of IFN-γ as well as IL-4, that is, both Th1 and Th2 cytokines [13]. This unique function of NKT cells is in sharp contrast to the function of conventional T cells, which detect antigens (mainly peptides) presented on MHC molecules, and are comprised of two functionally distinct subsets, namely, Th1 and Th2 cells.

When these cells are stimulated by cytokines or bacterial components, such as lipopolysaccharide (LPS), superantigens of gram-positive bacteria, or streptococcal derivatives, they play important roles in the defense against tumors and infections; i.e., both NKT and NK cells exert cytotoxic activity simultaneously or differentially by releasing cytotoxic mediators, such as perforin and granzyme, and/or by producing proinflammatory cytokines, including interferon-gamma (IFN-γ), depending on the type of stimuli [10]. On the other hand, inappropriate overactivation of NKT cells can cause shock or multiple organ failure (MOF), including injury of the liver, lung, and kidney, via the TNF-α/Fas ligand (FasL) pathway [15]. Activated NK cells have also been shown to exert cytotoxic activity against NK-resistant tumors expressing MHC class I molecules [16]. Thus, these innate immune cells can attack normal cells, if they are inappropriately overactivated. In line with these findings, their involvement in the pathogenesis of AKI through the damage to intrinsic renal cells has recently become evident [7].

Human NK cells are traditionally defined as CD56^+^ CD3^−^ large granular lymphocytes, and T cells expressing CD56 (CD56^+^ T cells), a surface marker of NK cells, as well as intermediate and oligoclonal TCRs, are considered to be human NKT cells [10,16,17]. Most CD56^+^ T cells show the CD8^+^ or CD4^−^CD8^−^ phenotype. Some researchers have proposed that human T cells expressing TCRs encoded by the Vα24Jα18 and Vβ11 genes, in which the arrangement of TCRs resembles that of mouse invariant NKT cells, are human NKT cells [18]. Indeed, these cells are activated by a specific sphingoglycolipid ligand of mouse invariant NKT cells [8], and their involvement in the pathophysiology of human diseases, including microscopic polyangiitis, a form of vasculitis that often causes damage to kidneys, was suggested [19]; however, these cells only constitute a very rare population both in the peripheral blood and liver in humans [14]. On the other hand, CD56^+^ T cells are a functional counterpart of mouse NKT cells in that (i) they are abundant in the liver, (ii) they show antitumor effects after cytokine stimulation, and (iii) most liver CD56^+^ T cells express CD161, which is identical to the NK1.1 antigen in mice. It should be noted, however, that although mouse NKT cells produce both Th1 and Th2 cytokines, human CD56^+^ T cells do not produce IL-4 and may not induce Th2 immune responses [20,21]. Nonetheless, human CD56^+^ T cells can produce proinflammatory cytokines after various types of stimulation, including infectious pathogens [20], and larger amounts of perforin and soluble FasL than conventional CD8^+^ T cells [22], and can damage normal cells that express MHC class I molecules, because human CD56^+^ T cells do not express the MHC class I inhibitory receptor CD94/NKG2A [7]. Importantly, it has recently been reported that activated CD56^+^ T cells (and a subpopulation of CD56^+^ NK cells) play crucial roles in the processes that mediate AKI by damaging intrinsic renal cells, as with murine NKT (and NK) cells [8].

In this review, we aim to introduce our current understanding of the association among infections, NKT and NK cells, and kidney injury, which is much more profound than previously thought. The roles of two types of liver macrophages in the activation/regulation of NKT cells are also discussed.

## 2. Roles of Mouse NKT Cells and NK Cells in Infection-Associated Kidney Injury

The proportion of both NKT cells and NK cells among total lymphocytes in the normal kidney are higher than those of the spleen and blood, suggesting the important roles of the innate immune response in the kidney. In addition, the proportion of NKT cells in the kidney that express CD69, an activation marker, increases with age [7].

Useful animal models of infection-associated renal diseases have not yet been established, because it is more difficult for animals to become infected by pathogens than humans. Therefore, the functions of mouse NKT cells have been investigated using α-galactosylceramide (α-GalCer), a synthetic sphingoglycolipid ligand of NKT cells. However, it was shown that although α-GalCer activates mouse NKT cells and induces antitumor responses that are mediated by NK cells (and subsequently CD8^+^ T cells), activated NKT cells also cause MOF via the TNF-α/FasL system, particularly in aged mice. Our group recently showed that NKT cells activated by α-GalCer cause AKI with hematuria, by injuring both renal vascular endothelial cells and tubular epithelial cells. The perforin-mediated pathway is involved in vascular endothelial cell injury, whereas the TNF-α/FasL system is mainly involved in the injury of tubular epithelial cells, and the two mechanisms play mutually independent roles. In addition, the function of NKT cells was enhanced in this model when NK cells were pharmacologically depleted. Not only α-GalCer, a synthetic reagent, but also some microbes have been proposed as antigenic targets of NKT cells, and CpG oligodeoxynucleotides (CpG-ODN), which are bacterial DNA motifs, also activate NKT cells and cause acute renal tubular injury via the TNF-α/FasL pathway [23]. Importantly, it was suggested in this model that NK cells interact with NKT cells and affect their deleterious function, which is in sharp contrast to the finding observed in the α-GalCer injection mouse model. In any case, it is reasonable to consider that NKT cells and NK cells play important roles in the disease progression of AKI in response to various microbes (Figure 1). Whether FasL-induced tubular epithelial cell injury starts in the urinary space by cells that rupture the glomerular capillary walls, or from the tubulointerstitial area by cells that extravasate from the injured peritubular capillaries, remains unclear. Furthermore, whether the origin of these innate immune cells are kidney-resident cells or cells recruited from other tissues, such as the liver, is another important issue that requires further study.

Renal ischemia-reperfusion injury (IRI) models are well-known experimental models of AKI, and the exacerbating roles of NKT cells during renal IRI have been demonstrated. It has been reported that the recruitment of NKT cells and the production of cytokines, such as IFN-γ and IL-17 by these cells lead to tissue damage in renal IRI [24], and involvement of the FasL pathway in the process of renal tubular injury in these models has been suggested [25]. The upregulated expression of a NK cell ligand on tubular epithelial cells and its engagement via the activating receptor on NK cells has also been reported, which also suggest the pathogenic roles of NK cells in renal IRI [26]. On the other hand, the protective roles of NKT cells in renal IRI have also been reported [27], and differences in the severity of renal injury might affect NKT cell function [9].

In the MOF model induced by IL-12 priming and subsequent LPS challenge (i.e., the generalized Shwartzman reaction, which is regarded as an exaggerated form of the response of hosts to microbes), both NKT cells and NK cells have been reported to play crucial roles, presumably via the perforin-mediated pathway. Thus, the functions of these innate immune cells appear to be double-edged swords, and it should be stressed that aging attenuates both antitumor and antimicrobial functions but aggravates tissue damage [16]. On the other hand, NKT cells have been shown to accelerate liver regeneration after hepatectomy via the TNF-α/FasL pathway [28], and it should be carefully investigated in the future as to whether they also play beneficial roles in the convalescence phase of infection-associated AKI.

## 3. Associations among Infections, Human CD56^+^ T Cells and CD56^+^ NK Cells, and Renal Diseases

Healthy kidneys have been reported to contain a large number of CD56^+^ NK cells (up to about one-quarter of total lymphocytes), including both CD56^dim^ and CD56^bright^ NK cells [12]. CD56^dim^ NK cells are the dominant subset in peripheral blood, whereas CD56^bright^ NK cells are the major NK cells in secondary lymphoid and peripheral tissues. Although CD16^−^ CD56^bright^ cells do not show cytotoxic effects under normal circumstances, when activated by several cytokines, some of them begin to express CD16, produce large amounts of IFN-γ, and exert strong antitumor functions against not only MHC class I-negative but also MHC class I-positive cells [29]. Interestingly, these CD16^+^ CD56^bright^ cells are reportedly induced by streptococcal derivatives [29]. CD56^bright^ NK cells are known to frequently express a marker of tissue residency in healthy kidneys, and it is possible that they play crucial functions in the defense against infections. CD16a (FcγRIII)-mediated recognition of the Fc portion of antibodies may also be involved. On the other hand, the overactivation of CD56^bright^ NK cells may cause organ damage. Indeed, in our recent study using peripheral blood mononuclear cells (MNCs) stimulated by IL-2, IL-12, and IL-15, we showed that CD56^bright^ cells substantially injure renal tubular epithelial cells in vitro; the cytotoxicity of CD56^bright^ NK cells was significantly higher than that of CD56^dim^ cells and conventional T cells [8]. Although limited data are available regarding the involvement of CD56^+^ T cells in renal diseases, cytokine-stimulated CD56^+^ T cells also exert strong cytotoxicity in vitro against intrinsic renal cells, such as glomerular endothelial cells and tubular epithelial cells. Importantly, glomerular endothelial cell injury was attenuated by inhibition of the perforin-mediated pathway, and it is suggested that a common pathogenic pathway is involved in both mouse and human conditions.

IgA nephropathy (IgAN) is a representative type of chronic glomerulonephritis accompanying inflammatory cell infiltration, and is a major cause of ESKD. Synpharyngitic gross hematuria is a well-known clinical syndrome that is thought to be associated with the exacerbation of glomerular vasculitis, and the potential therapeutic effect of tonsillectomy for IgAN has become widely recognized [30]. Interestingly, urinary MNCs of patients with IgAN contain significantly more CD56^+^ cells than peripheral blood MNCs [31], and CD56^+^ cells are found predominantly within the crescentic lesions of renal biopsy tissues [32]. In addition, both the proportion of CD56^+^ T cells and that of CD56^+^ NK cells in blood, as well as the expression of a chemokine receptor that is involved in the cytotoxic function of these cells, are reportedly increased during gross hematuria episodes [33]. Furthermore, the CD56^+^ cell has been shown to exert cytotoxicity against glomerular endothelial cells and to induce hematuria [34]. Thus, these cells are assumed to be involved in the exacerbation of pre-existing glomerulonephritis, which is triggered by infection. Some patients experience synpharyngitic gross hematuria even after undergoing a tonsillectomy. Immune responses in nasopharynx-associated lymphoid tissue, particularly in the epipharynx, are suggested to play important roles in this condition, and the concept of the epipharynx-kidney axis has recently been proposed [35]. Indeed, in murine IgAN, nasal immunization with CpG-ODN was shown to exacerbate kidney injury, whereas that by oral gavage did not [36]. Future studies investigating the detailed mechanism of the epipharynx-kidney axis and its association with innate immune cells is needed.

Infection-related glomerulonephritis (IRGN) is an immune-mediated glomerulonephritis caused by nonrenal infections. Patients with IRGN, particularly older patients with immunocompromised backgrounds, have a guarded prognosis [37,38]. Glomerular endocapillary proliferative and exudative glomerulonephritis with the massive infiltration of neutrophils and/or macrophages is a typical pathological finding, and glomerular deposition of nephritis-associated plasmin receptor and associated plasmin activity has been suggested as both diagnostic and pathological biomarkers [39,40]. We recently performed immunohistochemical staining for CD56 using renal biopsy tissues of patients with IRGN, and only found a small amount of CD56-positive cell infiltration (unpublished observation). Thus, the role of CD56^+^ T cells and/or NK cells in the pathogenesis of IRGN remains unclear to date. However, it should be stressed that there is a possibility that the number of these cells in the kidney does not have a significant effect; although NK cells are now considered as a key player in the pathogenesis of the antibody-mediated rejection of transplanted kidneys, they only constitute a rare population among infiltrating immune cells in renal biopsy tissues [41,42]. In addition, there is another possibility that these cells are involved through indirect mechanisms, such as the production of proinflammatory cytokines or proapoptotic molecules.

Effective therapeutic regimens for infection-associated AKI have not yet been established; however, inhibiting the overactivation of CD56^+^ T cells and/or NK cells appears to be a promising approach. From this point of view, targeting the perforin and/or FasL pathways would be reasonable. Indeed, acute inflammation as well as renal tubular injury, which is possibly caused by overactivation of the FasL pathway, have progressively emerged as pathogenetic mechanisms of sepsis-induced AKI [2]. The enhancement of bactericidal activity by phagocytic macrophages without causing an increase in the overproduction of proinflammatory cytokines is also an attractive strategy against AKI, and C-reactive protein (CRP), an acute-phase protein that is produced in response to infection, inflammation, or organ damage, and is widely used as a marker of the inflammatory response, may be a candidate therapeutic molecule with such functions. It has been reported that the treatment of mice with severe sepsis using a synthetic CRP molecule mitigates their liver injury and increases their survival rate [43,44]. Synthetic CRP treatment also suppressed the production of TNF-α and IL-12 from human peripheral blood MNCs that were stimulated in vitro with bacterial reagents, such as CpG-ODN or LPS, whereas it upregulated the functions of CD56^+^ NK cells [45]. Further studies investigating the effects of CRP treatment on sepsis-associated AKI, focusing on CD56^+^ NK cells and CD56^+^ T cells are anticipated in the future. Alkaline phosphatase (ALP), an endogenous enzyme expressed throughout the entire body, is another potent therapeutic molecule for sepsis-associated AKI [46]. ALP was shown to dephosphorylate and detoxify harmful molecules, including LPS, and to thereby inhibit inflammatory responses. Although the efficacy of ALP has recently been reported [47], and a new global clinical trial is currently underway, the actions of ALP on CD56^+^ NK cells and CD56^+^ T cells is a subject of future research.

We observed that intracellular perforin expression both on CD56^+^ NK cells and CD56^+^ T cells was increased in patients with sepsis-associated AKI who received continuous renal replacement therapy (CRRT). In contrast, in these patients, FasL expression was upregulated on CD56^+^ T cells, but not on CD56^+^ NK cells (Figure 2, unpublished preliminary data of the authors). The concentration of both serum and urine FasL was reportedly increased during the early disease stage in a sepsis-associated AKI model [48], and previous studies, including ours, demonstrated the important roles of the TNF-α/FasL system in the tubular epithelial cell injury of murine AKI models [7,23]. Therefore, it is highly likely that the FasL pathway is also involved in the pathogenesis of sepsis-associated AKI in humans. The proportion of NK cells was reportedly increased in patients after CRRT [49], whereas our analysis showed that the proportion of both CD56^+^ NK cells and CD56^+^ T cells of sepsis-associated AKI patients did not significantly differ from that of healthy controls [22]. Thus, the roles of NK cells as well as CD56^+^ T cells in human sepsis-associated AKI remain to be clarified, and further detailed studies are required.

## 4. Two Distinct Macrophage Subsets in the Liver and Kidney and Their Interactions with NKT Cells

We previously found that two macrophage subsets exist in the liver and the kidney [50,51,52,53]. In the liver, yolk sac- and fetal liver-derived resident macrophages (Kupffer cells), and macrophages derived from bone marrow monocytes (recruited macrophages) are present, and cooperate with each other. CD11b^low^ F4/80^high^ Kupffer cells are radioresistant and exert potent phagocytic and cytotoxic activity via the production of ROS against bacteria, as well as the endocytosis/digestion of lipoproteins. On the other hand, CD11b^high^ F4/80^low^ recruited macrophages are radiosensitive and produce cytokines, such as IL-12 and TNF-α. IL-12 produced by the recruited macrophages activate liver NKT cells and induce potent antitumor functions, not only in the liver but also in the lung as well as in the kidney [10,16]. Although TNF-α produced by these cells activates NKT cells to produce and express FasL, inducing hepatocyte injury, as described above, NKT cells accelerate hepatocyte regeneration after partial hepatectomy using the same TNF-α/FasL pathway [15,28,54,55]. Therefore, we propose that NKT cells attack and remove injured or infected hepatocytes but help newly generated hepatocyte growth, thereby maintaining hepatocyte homeostasis. A typical experimental model of the cooperation among Kupffer cells, recruited macrophages, and NKT cells is concanavalin A-induced acute hepatitis [56], in which ROS produced by Kupffer cells is a final hepatotoxic effector, and recruited macrophages that produce TNF-α as well as NKT cells that produce IFN-γ are required for the final activation of Kupffer cells and their ROS production. These two types of macrophages, namely, CD11b^low^ F4/80^high^ cells and CD11b^high^ F4/80^low^ cells, are also present in the kidney [53]; the former have potent phagocytic activity and the latter have potent TNF-α-producing capacity. Therefore, recruited macrophages may be involved in renal injury by producing TNF-α and activating NKT/NK cells in the kidney after bacterial infections.

Interestingly, we recently found that the nuclear receptor liver X receptor (LXR; its ligand is the cholesterol derivative oxysterol), contrastingly regulates the functions of Kupffer cells and recruited macrophages in the liver [52]. LXR stimulation by a synthetic ligand enhances the phagocytosis of bacteria and lipoproteins by Kupffer cells, but contrastingly inhibits the production of TNF-α by the recruited macrophages [52]. Consistently, LXR-knockout (KO) mice were found to be susceptible to bacterial infection owing to the impaired phagocytic activity of Kupffer cells and enhanced TNF-α production of the recruited macrophages [52,57]. Intriguingly, LXR KO mice showed a reduced number of Kupffer cells and quite a few NKT cells but an increased number of recruited macrophages in the liver [52,58]. Therefore, LXR may regulate the development of both liver macrophages and NKT cells and control bactericidal activity, cytokine production, cholesterol metabolism, and antitumor immunity, thereby maintaining the homeostasis of immunity and lipid metabolism in the liver. Future studies are required to investigate how NKT cells interact with Kupffer cells in detail, and how LXR affects the function of the two types of macrophages in the kidney.

## 5. Concluding Remarks

In addition to the already-known antitumor and antimicrobial functions, it has recently become evident that NKT and NK cells play significant roles in various renal diseases. We herein provided an overview of the recent advances in the understanding of the association between infections, NKT and NK cells, and kidney injury (Table 1).

The inappropriate activation of murine NKT cells by various stimuli, including bacterial components, can cause shock or MOF, including AKI. Both renal vascular endothelial cell injury via the perforin-mediated pathway and tubular epithelial cell injury via the TNF-α/FasL system are independently involved in these conditions. Mouse NK cells complement the functions of NKT cells and play pivotal roles in infection-associated AKI. Upon activation, human CD56^+^ T cells, which act as NKT cells, injure intrinsic renal cells in vitro, and there may be a common pathogenic pathway between mice and humans. A subpopulation of human CD56^+^ NK cells, namely, CD56^bright^ NK cells, also cause substantial damage to intrinsic renal cells. Thus, it is assumed that these cells are involved in the processes that mediate AKI, as with mouse NKT and NK cells.

Infections often trigger the aggravation of pre-existing glomerulonephritis, such as IgAN, and human CD56^+^ NK cells as well as CD56^+^ T cells are thought to play important roles in this phenomenon. The involvement of these cells in the pathogenesis of human sepsis-associated AKI is currently being investigated, but it is interesting that a therapeutic strategy that modulates innate immune responses has shown favorable results in the field of both basic and clinical research. The expression of cytotoxic molecules on CD56^+^ NK cells and CD56^+^ T cells appears to be increased in patients with severe sepsis-associated AKI, and the effects of inhibiting the overactivation of these cells is a subject of future research.

The interactions among murine two macrophage subsets, namely, CD11b^low^ F4/80^high^ cells and CD11b^high^ F4/80^low^ cells, NKT cells, and NK cells in the kidney injury should be further investigated in future studies, because the findings would shed light on the roles of these cells in the development of infection-associated kidney injury in humans.

## Figures and Tables

**Figure 1 ijms-23-00479-f001:**
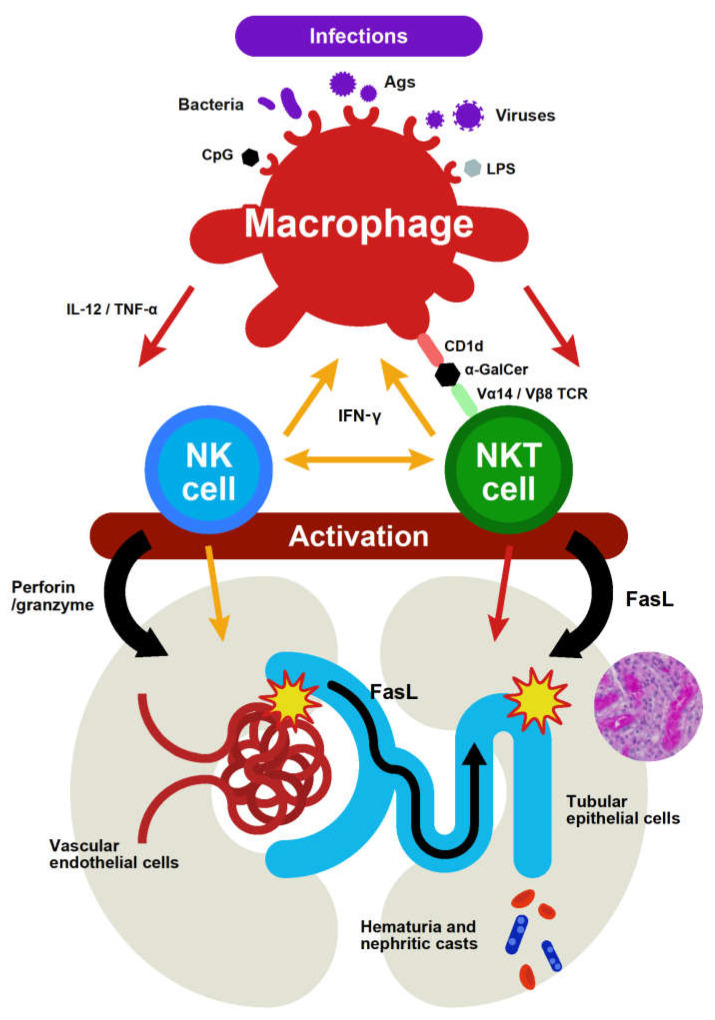
Putative pathogenic mechanisms of AKI induced by mouse NKT cells and NK cells in response to various stimuli, such as their specific ligand or bacterial components.

**Figure 2 ijms-23-00479-f002:**
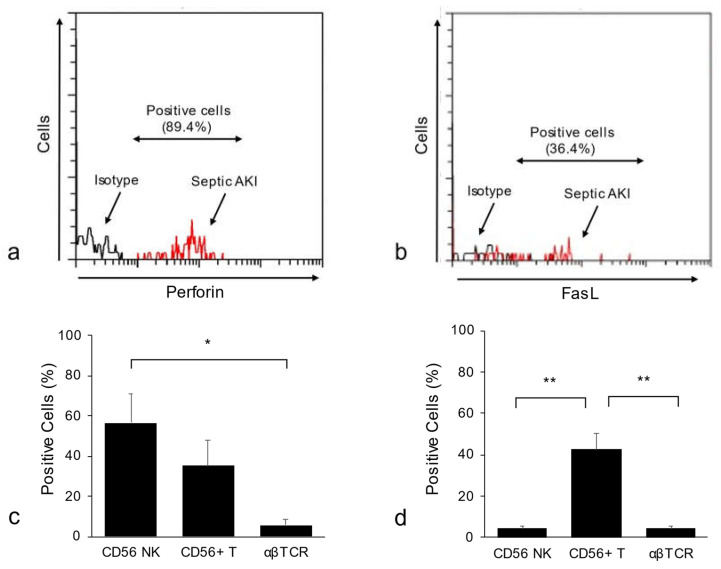
Proportion of peripheral blood MNCs of patients with sepsis-associated AKI receiving continuous renal replacement therapy and who are positive for cytotoxic effector molecules. Representative histogram showing perforin expression on CD56 NK cells (**a**) and FasL expression on CD56^+^ T cells (**b**). The percentages of perforin-positive cells (**c**) and FasL-positive cells (**d**) (*n* = 5 in each group). Data are presented as the means ± SE. * *p* < 0.05, ** *p* < 0.01. (Unpublished preliminary data of the authors).

**Table 1 ijms-23-00479-t001:** Relation between infections, reactions of NKT cells and NK cells, and kidney injury.

Causative Pathogens	Roles of NKT and/or NK Cells	Type of Kidney Injury
CpG-ODN *^1^ [23]	Tubular epithelial cell injury via the TNF-α/FasL system	AKI in mice
LPS (the generalized Shwartzman reaction) [8]	Renal vascular endothelial cell injury via the perforin-mediated pathway	
Certain bacteria	Increase in expression of perforin and FasL	Sepsis-associated AKI in human
CpG-ODN *^2^ [36]	Mechanism remains to be solved	Exacerbation of murine IgAN
Certain bacteria	Injury of glomerular endothelial cells leading to rupture of capillary walls	Exacerbation of human IgAN

*^1^ intravenous injection, *^2^ nasal immunization.

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
