# Peer review of "Infections, Reactions of Natural Killer T Cells and Natural Killer Cells, and Kidney Injury"

_ijms, 2022, doi:10.3390/ijms23010479_

Round 1
Reviewer 1 Report
Uchida et al. tried to explore in detail NKT and NK cells in the context of inflammatory state, emphasizing the role of kidney injury in this field. It appears to me that the Authors presented already known data concerning this subject in a very comprehensive way, indeed. Additionally, they referred to their own experience from conducted studies on these issues - it is also a strong point of this manuscript. The article sounds like a review, however it is closely attached to the clinical perspective. The only one thing I would like to ask the Authors for is creating an additional paragraph, closing the manuscript, which would focus on the future perspective for examined cell subsets in general and in case of kidney injury.
Author Response
We sincerely thank you for your kind and positive comments and a suggestion. Our response is shown below.
The only one thing I would like to ask the Authors for is creating an additional paragraph, closing the manuscript, which would focus on the future perspective for examined cell subsets in general and in case of kidney injury.
Response:
We added a paragraph describing the future perspective on these cells in kidney injury in the final part of the manuscript as advised (Page 9, lines 12-15).
Reviewer 2 Report
An article submitted for review summarizes the current state of knowledge in an association among infections, NKT and NK cells, and kidney injury. The review article integrates and interprets previous results of original research studies. In addition, it includes the authors' original research findings. The references selection is sound and up-to-date. With the suggested corrections, I recommend the article for publication.
The manuscript is well written and justified through suitable references. It contains sufficient novelty to be accepted for publication. I suggest that the authors include a table summarising the main theses presented in the article.
Author Response
We sincerely thank you for your kind and positive comments and a suggestion. Our response is shown below.
I suggest that the authors include a table summarising the main theses presented in the article.
Response:
We made a table summarizing the main theses presented in our article as advised (Table 1 in Page 8).